# Chlamydia Peritonitis Mimicking Juvenile Carcinomatous Peritonitis Diagnosed by Exploratory Laparoscopy: A Case Report

**DOI:** 10.3390/pathogens12010094

**Published:** 2023-01-06

**Authors:** Haruka Nishida, Yuko Takahashi, Kohei Takehara, Keita Yatsuki, Takayuki Ichinose, Tsuyoshi Ishida, Haruko Hiraike, Yuko Sasajima, Kazunori Nagasaka

**Affiliations:** 1Department of Obstetrics and Gynecology, Teikyo University School of Medicine, Tokyo 173-8605, Japan; 2Department of Pathology, Teikyo University School of Medicine, Tokyo 173-8605, Japan

**Keywords:** *Chlamydia trachomatis* infections, ovarian cancer, exploratory laparoscopy

## Abstract

*Chlamydia trachomatis* infections may occur in multiple organs, including the lungs, lymph nodes, peritoneal cavity, and genitourinary systems. This disease results in significant ascites, the swelling of lymph nodes, and elevated tumor markers (CA125), sometimes mimicking an ovarian malignancy. At our hospital, we often perform examination laparoscopic surgery in cases of suspected gynecologic cancers before initial treatment. In this paper, we report the case of a 19-year-old woman who came to our hospital because of an ovarian tumor and ascites. There was no history of sexual intercourse (self-reported). We suspected ovarian cancer from image inspections, so we performed laparoscopic surgery for diagnosis. The final pathological diagnosis was acute-to-chronic inflammation of the bilateral fallopian tubes, and a cytologic examination of the ascites was negative for malignant cells. The *C. trachomatis* antigen was positive on vaginal examination after the operation. Based on this result, we diagnosed this patient with *C. trachomatis* infection. Chlamydia peritonitis should be a differential diagnosis for cancer peritonitis in juvenile patients with abnormal ascites. Exploratory laparoscopy should help confirm the pathological diagnosis.

## 1. Introduction

Chlamydia infection is the most common sexually transmitted disease (STD) in Japan [1]. Reporting STDs is mandatory for gonococcal infection, genital herpes virus infection, and condyloma acuminatum. Still, a review of notifications since the Infectious Diseases Act came into force shows a sharp increase in the number of female patients reported owing partly to the rise in the number of fixed points for obstetrics and gynecology under the Infectious Diseases Control Act compared to that under the old surveillance. Nevertheless, the number of infected female patients is increasing [1]. As 3–5% of pregnant women are found to have chlamydial infection during antenatal check-ups, there are many infected women without any subjective symptoms [2]. Adults are infected via the sexual transmission of *Chlamydia trachomatis*, while newborns are infected through the birth canal by their mothers. Chlamydia infection is common in sexually active young people of both sexes. Recently, the high infection rate among teenage girls has been of concern because of a decline in the age of first sexual intercourse, which may lead to future infertility. Attention should be paid to the fact that women are often unaware of the infection and, therefore, do not receive diagnostic treatment. Further, they are unaware of the symptoms that may lead to transmission to the male partner or the child during childbirth. Infection of the pharynx due to oral intercourse is not uncommon.

*C. trachomatis* infections may occur in multiple organs, including the lungs, lymph nodes, peritoneal cavity, and genitourinary systems, among young women. Pelvic inflammatory disease (PID) due to *C. trachomatis* may result in significant ascites, pelvic masses, cul-de-sac nodularity on an exam, elevated levels of the serum tumor marker CA125, and imaging findings that mimic ovarian malignancy [3].

At our hospital, we often perform examination laparoscopic surgery in cases of suspected gynecologic cancers before initial treatment. One of the advantages of examination laparoscopic surgery for suspected gynecologic cancer is the confirmation of the diagnosis following tissue sampling. With advances in imaging, it is possible to estimate the possibility of cancerous peritonitis preoperatively in many cases; however, it is also known that peritoneal tuberculosis and malignant mesothelioma can present with similar imaging findings. This time, we performed examination laparoscopic surgery in a case of suspected gynecologic cancer, and the pathological diagnosis was oviductitis, which was eventually diagnosed as chlamydia infection.

## 2. Case Report

A 19-year-old female patient (0 gravidae) with no history of sexual intercourse (self-reported) visited our hospital. She reported a history of Kawasaki disease from the age of five years old. She had a regular menstrual cycle. The patient was referred to our hospital for the first time because of a 4 cm right ovarian tumor with ascites. At her first visit to our hospital, she had a hematoma or right ovarian tumor suspected of having a solid component and ascites. Her abdomen was soft and flat with no tenderness. Except for D-dimer levels of 7.4 μg/mL, C-reactive protein (CRP) levels of 1.02 mg/dL, and CA125 levels elevated to 923.9 U/mL, her blood test results were consistent. *C. trachomatis* and *Neisseria gonorrhoeae* DNA probe tests were not performed. We suspected ovarian cancer and arranged for an image inspection. Because of the evaluation of D-dimer levels, we performed ultrasonography to diagnose venous thromboembolism in the lower limbs, but thrombosis was not detected. Magnetic resonance imaging (MRI) showed a suspected right hemorrhagic ovarian cyst and massive ascites (Figure 1A). A computed tomography (CT) scan showed massive ascites and peritoneal thickening, with suspected peritoneal dissemination (Figure 1B). Positron emission tomography (PET) showed para-aortic lymph node metastasis, peritoneal dissemination, and liver metastasis. No significant accumulation was found in the uterus or bilateral adnexa (Figure 2). The upper and lower gastrointestinal endoscopies showed normal findings. We suspected gynecologic cancer, and one month after her first visit to our hospital, laparoscopic examination surgery was scheduled. She had abdominal pain and was hospitalized four days before surgery. The findings on the admission of an ovarian cyst were unclear, and the severity of ascites was unchanged on a transrectal ultrasound. She had no fever, and her blood test results had not changed remarkably-the D-dimer levels were 6.6 μg/mL, the CRP level was 0.18 mg/dL, and the CA125 level was elevated to 923.9 U/mL. An abdominal radiography showed no abnormalities. The causes of the worsening abdominal pain were unclear; after admission, reasonable pain control was achieved with non-steroidal anti-inflammatory drugs (NSAIDs) and acetaminophen medication. We performed the laparoscopic examination surgery as scheduled. The intraoperative findings showed about 200 mL of viscous yellow ascites and inflammatory peritoneum. There were some white lesions similar to intraperitoneal dissemination. The uterus was inflamed and adhered to the Douglas fossa, and the bilateral fallopian tubes were swollen. We considered fallopian tube cancer (Figure 3). There were no abnormal findings on the liver surface, where a disseminated lesion was suspected based on the preoperative imaging examination, including adhesion to the peritoneum. The tissues around the right fallopian tube and peritoneum were biopsied and submitted for pathological examination. In the biochemical test of ascites, adenosine deaminase (ADA), hyaluronic acid, and CA125 levels increased remarkably; the ADA level was 94.3 U/L, the hyaluronic acid level was 47,110 ng/mL, and the serum CA125 level further elevated to 2409.2 U/mL. The bacterial examination was negative. Therefore, we further suspected tuberculous peritonitis; however, there was no evidence of any significant clinical symptoms for the diagnosis of tuberculosis. She was discharged 3 days after surgery without any exacerbation of abdominal pain. The final pathological diagnosis was acute-to-chronic inflammation of the bilateral fallopian tubes, and a cytologic examination of ascites was negative for malignant cells. The pathological images (HE staining) of the fimbrial fallopian tubes and Douglas fossa peritoneum biopsied at the time of surgery are shown in Figure 4 and Figure 5. The result of the bacterial test of the vaginal discharge revealed *Lactobacillus species*+ and *Staphylococcus species*+. A vaginal examination revealed positive *C. trachomatis* antigen, and the test for the acid-fast bacterium was negative. Thus, we considered that the cause of this case was *C. trachomatis* infection. We conducted a *C. trachomatis* DNA probe test two weeks after her hospital discharge, and the results were positive. We always collect endocervical mucus with a cotton swab; store it in cobas^®^ PCR media (Roche Tokyo, Japan); submit it to SRL, Inc. (SRL Tokyo, Japan); and perform a chlamydia PCR test. We administered azithromycin as treatment. Two weeks after starting the azithromycin treatment, the *C. trachomatis* DNA probe test was negative, and the value of CA125 decreased to 24.2 U/mL. We performed a transrectal ultrasound and confirmed the disappearance of the ascites. Her abdominal pain also diminished. The graphs of the values of serum CA125 are shown in Figure 6, those of the white blood cells are shown in Figure 7, and those of CRP are shown in Figure 8; day 0 means the date of the first visit to our hospital.

## 3. Discussion

*C. trachomatis* infections may occur in multiple organs, including the lungs, lymph nodes, peritoneal cavity, and genitourinary systems. Therefore, CT and PET-CT examinations may resemble ovarian malignancies [3]. In this case, *C. trachomatis* and *N. gonorrhoeae* DNA probe tests were not performed because the patient said that she had no history of sexual intercourse. In Japan, in 2020, the prevalence of chlamydia infection in women aged 15 to 19 years was 12%. There are reports that it is increasing at a young age. The most prevalent age groups are 20 to 24 years old, followed by 25 to 29 years old [1,4]. We excluded PID from the differential diagnosis, so we strongly suspected ovarian cancer. We should have suspected a sexually transmitted disease, regardless of whether the patient said that she had no history of sexual intercourse. A previous study has shown that young women do not want to be asked about their sexual activity before being tested for sexually transmitted diseases [5]. In the same study, some women said that they would lie if asked about their sexual activities. Therefore, it is advisable to check for sexually transmitted diseases without explicitly asking women about their sexual history, albeit informing them that they will be checked for sexually transmitted diseases.

We suspected tuberculous peritonitis from the biochemical test of ascites because the ADA and hyaluronic acid levels were remarkably elevated. There is a case report that the properties of ascites due to chlamydia infection and tuberculous ascites are similar [6], but it is difficult to diagnose from a test of ascites only. A chlamydia PCR test of ascites may also have been practical [6,7].

Despite the severe peritonitis caused by *C. trachomatis* infections, the patient had mild symptoms, including no fever and a slight elevation of WBC and CRP (Figure 7 and Figure 8). C. *trachomatis* has also been investigated in the villi and peripheral blood mononuclear cells (PBMCs) obtained from females with spontaneous abortion; however, the important point of this report was that all pregnant women were asymptomatic [8]. We should always suspect chlamydia infection even when examining asymptomatic women. The patient had abdominal pain and was hospitalized five days before surgery, but the episode of abdomen pain only occurred this one time. Chlamydia infection has been reported to have few symptoms, even after it has progressed to pelvic inflammation [9]. New testing methods, such as the testing of single-nucleotide polymorphisms (SNPs), which are thought to be involved in the immune response, as described in a previous report [9], may be desirable.

In our case, the high CA125 level was one factor for the suspicion of a malignant tumor. Previously, it has been reported that PID caused by chlamydia infection has a significantly higher CA125 value than PID caused by other infections. In the same study, patients with PID in the chlamydia group were younger and had a higher rate of TOA than the patients with PID in the non-chlamydia group. In addition, the erythrocyte sedimentation rate (ESR) and CRP level were higher in the chlamydia group than in the non-chlamydia group (the mean levels of the patients with chlamydia infection were ESR: 45.0 ± 26.6 mm/h, CRP: 7.6 ± 7.0 mg/dL, and CA125: 130.7 ± 174.6) [10]. In our case, ESR was not measured, the CRP level was not significantly elevated, and the CA125 level was clearly higher than that previously reported. However, of course, it is also important not to suspect a malignant tumor based on CA125 elevation alone.

Previously, there was a report that cancer can be caused by persistent chlamydia infection. There are several types: cervical dysplasia and cancer caused by C. *trachomatis*, lung cancer and cutaneous T-cell lymphoma caused by C. *pneumoniae,* and several non-gastrointestinal MALT lymphomas caused by C. *psittaci*, suggesting a potential role [11]. Moreover, there are reports of the identification of chlamydia from ocular lymphomas [12].

Finally, we administered azithromycin as treatment after the chlamydia infection was revealed by the PCR test. Based on the clinical course and pathological examination results of this case, the patient was considered to have an acute-to-chronic chlamydial infection. Although it is unclear how the patient’s chlamydial infection was cured during the long clinical course, considering that oral azithromycin treatment resulted in negative chlamydia PCR, it can be inferred that it at least contributed to the microbiological improvement.

## 4. Conclusions

Chlamydia peritonitis should be considered in the differential diagnosis of cancer peritonitis in juvenile patients with abnormal ascites. Exploratory laparoscopy should be helpful to confirm the pathological diagnosis.

## Figures and Tables

**Figure 1 pathogens-12-00094-f001:**
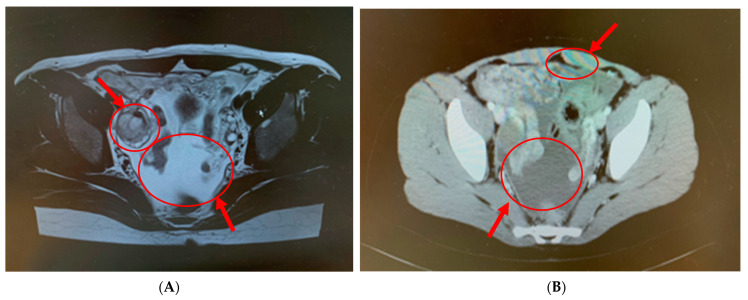
(**A**) MRI showed suspected right hemorrhagic ovarian cyst (left circle) and massive ascites (right circle). (**B**) Computed tomography (CT) scan showed massive ascites (lower circle), peritoneal thickening (upper circle), and suspected peritoneal dissemination.

**Figure 2 pathogens-12-00094-f002:**
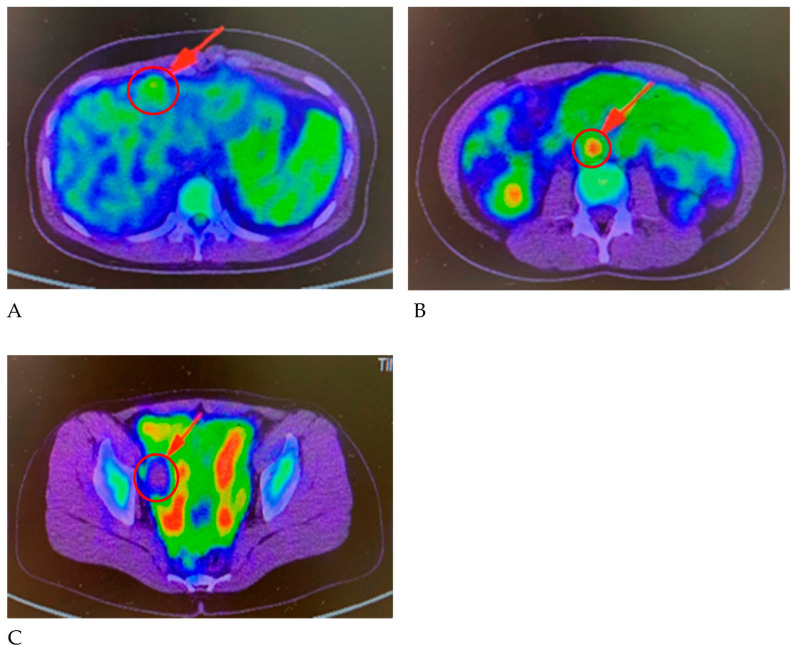
(**A**) PET showed liver metastasis, (**B**) para-aortic lymph node metastasis, and (**C**) peritoneal dissemination. No significant accumulation of 18F-FDG was found in the uterus or bilateral adnexa.

**Figure 3 pathogens-12-00094-f003:**
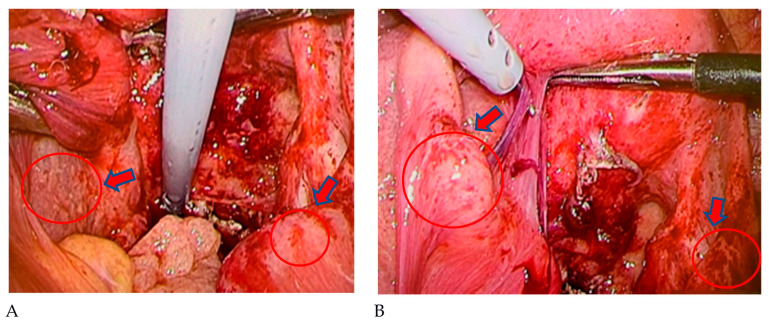
Intraoperative findings: (**A**) there were some white lesions similar to intraperitoneal dissemination (red arrow); (**B**) the uterus was inflammatory and adhered to the Douglas fossa, and the bilateral fallopian tubes were swollen (red arrow).

**Figure 4 pathogens-12-00094-f004:**
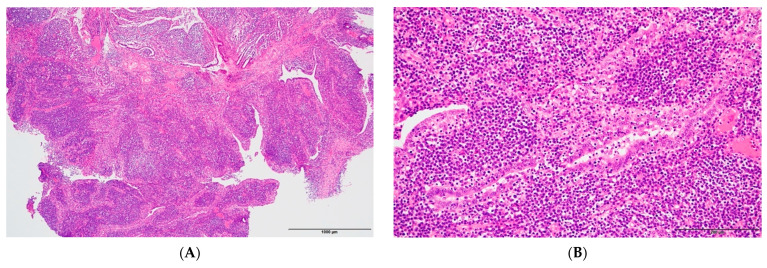
The images of HE staining of the right fimbrial fallopian tubes: (**A**) is high-power field, and (**B**) is low-power field. The wrinkle is markedly thickened, with a high degree of inflammatory cell infiltration. Inflammatory cells are lymphocytes, plasma cells, and neutrophils. There are no findings suggestive of malignancy.

**Figure 5 pathogens-12-00094-f005:**
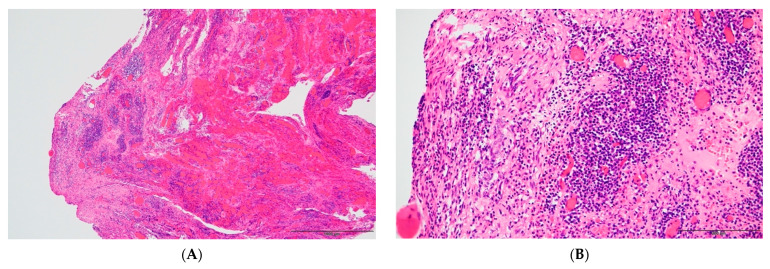
The images of HE staining of the Douglas fossa peritoneum: (**A**) is high-power field, and (**B**) is low-power field. Inflammatory cell infiltration, mainly composed of lymphocytes and plasma cells.

**Figure 6 pathogens-12-00094-f006:**
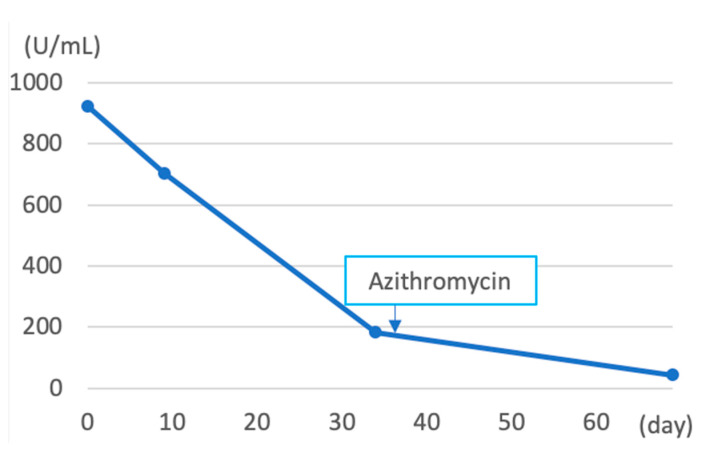
The value of serum CA125 level after the patient’s first visit to our hospital (day 0; the date of first visit). CA125 level was highest on the date of the first visit, at 923.9 U/mL. After the first visit, the value of CA125 decreased gradually.

**Figure 7 pathogens-12-00094-f007:**
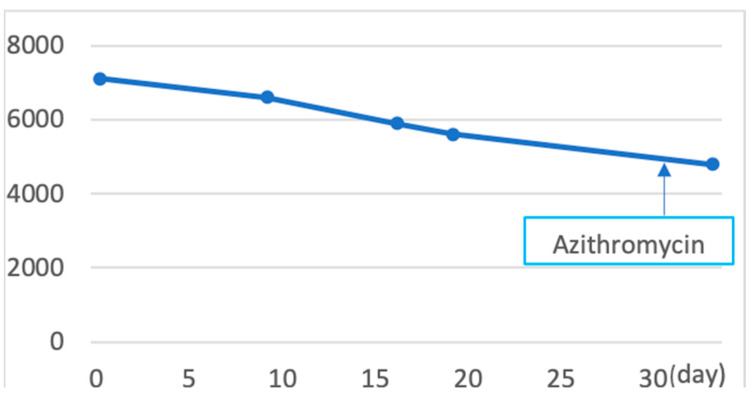
The value of white blood cells (WBCs) after the patient’s first visit to our hospital (day 0; the date of first visit). WBC value was highest on the date of the first visit, at 7100. After the first visit, the value of WBC decreased gradually.

**Figure 8 pathogens-12-00094-f008:**
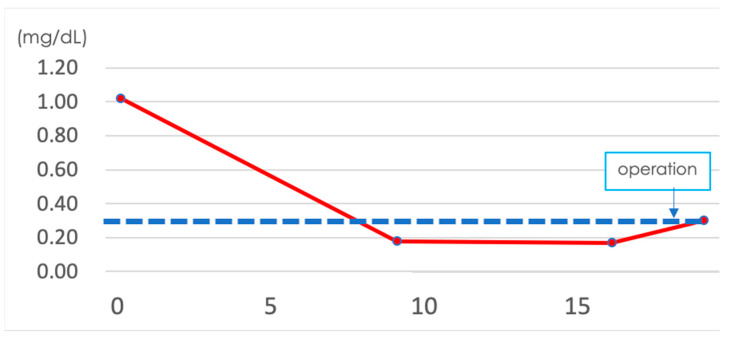
The value of CRP after the patient’s first visit to our hospital (day 0; the date of first visit). The dashed line indicates the normal ranges (CRP ≤ 0.3 mg/dL). Though we checked CRP only after operation, the value of CRP was highest on the date of the first visit, at 1.02 mg/dL. After the first visit, the value of CRP decreased gradually, and it was elevated slightly after operation.

## Data Availability

Not applicable.

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
