# Peer review of "Chlamydia Peritonitis Mimicking Juvenile Carcinomatous Peritonitis Diagnosed by Exploratory Laparoscopy: A Case Report"

_pathogens, 2023, doi:10.3390/pathogens12010094_

Round 1

Reviewer 1 Report (New Reviewer)

The authors describe chlamydial peritonitis mimicking ovarian cancer in a 19-yr old female patient. One of the main factors delaying the correct diagnosis was that the patient denied having had sexual intercourse, for which PID was not included in the differential diagnosis. Medical imaging and elevated CA12-5 antigen  referred towards ovarian tumour (although CA12-5 is known to be unspecific as a tumour marker). There is also some inconsistency between the text (line 77) and figure 5A, since CA12-5 concentration had decreased, not increased between the visits. As compared to the extensive medical imaging described, the laboratory methods used seem relatively oldfashioned: diagnosis of chlamydial infection was finally done by antigen test (brand or any further information not mentioned in the text), and probe tests were mentioned as alternatives. Why were modern molecular methods not used in the microbiological diagnostics? The authors conclude that chlamydial infection should be routinely included in differential diagnosis and tests performed irrespective of, or even without any sexual anamnesis. The manuscript is fluently written (although there is some repetition) and the figures are fine

Author Response

Dear Reviewer 1, 

Thank you very much for your suggestions. We respond to the points raised.

ーーーーーーーーーーー

The authors describe chlamydial peritonitis mimicking ovarian cancer in a 19-yr old female patient. One of the main factors delaying the correct diagnosis was that the patient denied having had sexual intercourse, for which PID was not included in the differential diagnosis. Medical imaging and elevated CA12-5 antigen referred towards ovarian tumour (although CA12-5 is known to be unspecific as a tumour marker).

→Thank you for your comments. As you pointed out, the serum CA125 value is as unspecific as the tumor marker. We suspected only the value of CA125 but also ascites and preoperative imaging. About CA125, we added a discussion (lines 201-210). Furthermore, we added references for CA125 and chlamydia infection; reference [11].

There is also some inconsistency between the text (line 77) and figure 5A, since CA12-5 concentration had decreased, not increased between the visits.

→Thank you for your comment. We corrected the value of CA125 at the patient’s first visit, line 80.

As compared to the extensive medical imaging described, the laboratory methods used seem relatively oldfashioned: diagnosis of chlamydial infection was finally done by antigen test (brand or any further information not mentioned in the text), and probe tests were mentioned as alternatives.

Why were modern molecular methods not used in the microbiological diagnostics? The authors conclude that chlamydial infection should be routinely included in differential diagnosis and tests performed irrespective of, or even without any sexual anamnesis. The manuscript is fluently written (although there is some repetition) and the figures are fine

→In clinical practice, we routinely diagnose Chlamydial infection by collecting patient cervical mucus and detecting Chlamydial PCR. The Chlamydial PCR is an outsourced examination to SRL in Japan, and the results will take several days. We added the inspection method and about media (lines 101-104). We should examine the Chlamydia PCR test if the patient said not to have sexual intercourse.

Reviewer 2 Report (New Reviewer)

I read the manuscript entitled "Chlamydia peritonitis mimicking juvenile carcinomatous peritonitis diagnosed by exploratory laparoscopy: A Case Report" by Haruka Nishida et al.

Criticism:

1. there are many other cancers sustained by C. trachomatis including those cited by the authors. Among these, there are  sites such as the ocular adnexa where C. trachomatis can cause ocular lymphomas (Contini C et al; Am J Haematol 2009; 84:597-9).  Moreover, please read and cite: Contini C. and Seraceni S. Chlamydial disease: a crossroad between chronic infection and development of Cancer. Bacterial and Cancer. (Chapter book, Springer Science Ed.) 2012, Chapter 4:79-116.

2. line 57-58: authors state that the young patient had no history of sexual intercourse (self reported) when visited at their hospital. However, C. trachomatisis may be transmitted through any type of sexual intercourse (vaginal, anal but also oral) and give a chronic infection which is often asymptomatic. Have the authors well investigated this way of transmission? Often, young women, as in the present case, deny having had sexual intercourse, but they may have had unprotected oral intercourse. It is estimated that about 70-80% of women and 50% of men are asymptomatic.

3. Introduction: the authors state that ….."this time, they performed a laparoscopic surgical examination on a suspected case of gynecological cancer, and the pathological diagnosis was oviductitis, which was eventually diagnosed as chlamydial infection". However, before laparoscopic surgery, Chlamydia should have been sought even without there having been previous sexual intercourse. And this, in light of the epidemiological data you cited and the rising incidence of C. trachomatis in Japan as well as worldwide (lane 33-34:...as 3–5% of pregnant women are found to have chlamydial infection during antenatal check-ups, there are many infected women without any subjective symptoms). Please, cite and read: Contini C, C e al. Investigation on silent bacterial infections in specimens from pregnant women affected by spontaneous miscarriage. J Cell Physiol. 2018 Jan;234(1):100-107. doi: 10.1002/jcp.26952. Epub 2018 Aug 5Also...line 157: in Japan, by 2020, the prevalence of 157 chlamydia infection in women aged 15 to 19 years was 12%....

5. the authors cite an antigen: which trachomatis antigen? Which manufacture? Which Country this antigen test did came from? Which significance? Did the authors trye to identify C. trachomatis by C. trachomatis DNA in the urine? C. trachomatis infection is diagnosed by molecular laboratory tests based on nucleic acid amplification (Naat), which, at present, due to their high sensitivity (>95%) and specificity (98%) are considered the reference tests for the diagnosis of chlamydia infection. These tests allow the detection of chlamydia in either endocervical and/or urethral swabs, vaginal, rectal, oral swabs, or urine samples. In case of positivity for chlamydia, it is recommended that the woman or man and partner(s) be serologically tested for HIV and tested for other sexually transmitted infections. The authors state that "We conducted a C. trachomatis DNA probe  two weeks after her hospital discharge, and the results came back positive".This test should have been done first...

6. Carbohydrate antigen 125 (Ca 125) is  specific and is highly confounding and should not be considered for cancer diagnosis, because it increases in any form of inflammation.

7. line 97-98. I find it hard to believe that two weeks after azithromycin administration, the C. trachomatis DNA test was negative. The C. trachomatis infection was already chronic although misrecognized, and C. trachomatis, like all Chlamydiae produce aberrant forms in their biological cycle that do not respond to therapy (Dreses-Werringloer U, Padubrin I, Jürgens-Saathoff B, et al (2000) Persistence of Chlamydia trachomatis is induced by ciprofloxacin and ofloxacin in vitro. Antimicrob Agents Chemother 44:3288-97).

Author Response

Dear Reviewer 2,

Thank you very much for your useful comments. Indeed, we have learned a lot from your suggestion. We have modified the text accordingly, and now we think the manuscript has been improved.

-------------------------------

Reviewer 2.

I read the manuscript entitled "Chlamydia peritonitis mimicking juvenile carcinomatous peritonitis diagnosed by exploratory laparoscopy: A Case Report" by Haruka Nishida et al.

Criticism:

  1. there are many other cancers sustained by C. trachomatis including those cited by the authors. Among these, there are  sites such as the ocular adnexa where C. trachomatis can cause ocular lymphomas (Contini C et al; Am J Haematol 2009; 84:597-9).  Moreover, please read and cite: Contini C. and Seraceni S. Chlamydial disease: a crossroad between chronic infection and development of Cancer. Bacterial and Cancer. (Chapter book, Springer Science Ed.) 2012, Chapter 4:79-116.

→ I read your comments, and I learned a lot. Thank you for your advice. We cited and added the discussion (lines 164-167.References [12]).

  1. line 57-58: authors state that the young patient had no history of sexual intercourse (self reported) when visited at their hospital. However, C. trachomatisis may be transmitted through any type of sexual intercourse (vaginal, anal but also oral) and give a chronic infection which is often asymptomatic. Have the authors well investigated this way of transmission? Often, young women, as in the present case, deny having had sexual intercourse, but they may have had unprotected oral intercourse. It is estimated that about 70-80% of women and 50% of men are asymptomatic.

→ We had not investigated the PCR of pharyngeal mucus. Pharyngeal infection is unlikely to cause peritonitis, so we think it was necessary to check the Chlamydia PCR of the cervical mucus.

  1. Introduction: the authors state that ….."this time, they performed a laparoscopic surgical examination on a suspected case of gynecological cancer, and the pathological diagnosis was oviductitis, which was eventually diagnosed as chlamydial infection". However, before laparoscopic surgery, Chlamydia should have been sought even without there having been previous sexual intercourse. And this, in light of the epidemiological data you cited and the rising incidence of C. trachomatis in Japan as well as worldwide (lane 33-34:...as 3–5% of pregnant women are found to have chlamydial infection during antenatal check-ups, there are many infected women without any subjective symptoms). Please, cite and read: Contini C, C e al. Investigation on silent bacterial infections in specimens from pregnant women affected by spontaneous miscarriage. J Cell Physiol. 2018 Jan;234(1):100-107. doi: 10.1002/jcp.26952. Epub 2018 Aug 5Also...line 157: in Japan, by 2020, the prevalence of 157 chlamydia infection in women aged 15 to 19 years was 12%....

→ As you pointed out, even if there was a complaint that there was no history of sexual intercourse, the test should have been done. I read the report and cited it in the discussion. (lines 168-175. References [8]).

  1. the authors cite an antigen: which trachomatis antigen? Which manufacture? Which Country this antigen test did came from? Which significance? Did the authors trye to identify C. trachomatis by C. trachomatis DNA in the urine? C. trachomatis infection is diagnosed by molecular laboratory tests based on nucleic acid amplification (Naat), which, at present, due to their high sensitivity (>95%) and specificity (98%) are considered the reference tests for the diagnosis of chlamydia infection. These tests allow the detection of chlamydia in either endocervical and/or urethral swabs, vaginal, rectal, oral swabs, or urine samples. In case of positivity for chlamydia, it is recommended that the woman or man and partner(s) be serologically tested for HIV and tested for other sexually transmitted infections. The authors state that "We conducted a C. trachomatis DNA probe  two weeks after her hospital discharge, and the results came back positive".This test should have been done first...

→Suspected chlamydia infection, we always collect endocervical mucus with a cotton swab. Moreover, we store it in cobas®ï¸Ž PCR media (the test kit from Roche). We added the details (lines 101-104). We did not try to identify C. trachomatis in the uterine. This test kit can also detect Chlamydia PCR from urine, so there would have been an option to test if there was resistance to transvaginal examination. She still insisted that she had never had sexual intercourse after I told her the results of the Chlamydia PCR test. So, I could not to talk about her partner’s treatment in detail. However, when sexually transmitted infections are found, we always tell them that their partner needs treatment. In addition, this patient was negative for HIV infection in the preoperative routine blood test (negative for HIV antibody and antigen).

  1. Carbohydrate antigen 125 (Ca 125) is  specific and is highly confounding and should not be considered for cancer diagnosis, because it increases in any form of inflammation.

→As you pointed out, CA125 is unspecific as a tumor marker. We added comments to the discussion and cited references (line201-210, references [11]).

  1. line 97-98. I find it hard to believe that two weeks after azithromycin administration, the C. trachomatis DNA test was negative. The C. trachomatis infection was already chronic although misrecognized, and C. trachomatis, like all Chlamydiae produce aberrant forms in their biological cycle that do not respond to therapy (Dreses-Werringloer U, Padubrin I, Jürgens-Saathoff B, et al (2000) Persistence of Chlamydia trachomatis is induced by ciprofloxacin and ofloxacin in vitro. Antimicrob Agents Chemother 44:3288-97).

→Pathological findings also indicated acute to chronic inflammation, so there is a possibility that the inflammation was chronic. However, we have not administered any antibiotics for her at our hospital (only 1g of cefazoline was administered prophylactically before surgery). After the administration of azithromycin as an outpatient, the result of the Chlamydia PCR test turned negative. Therefore, we think the azithromycin would have had a therapeutic effect. It has been a long time since we detected her massive ascites, although it is speculative, and there is a possibility that natural immunity may have contributed to her recovery. At her last medical examination, she had no abdominal symptoms, no ascites detected from a transvaginal ultrasound, and CA125 decreased to 24.2 U/mL (standard value of 36 U/mL or less). For these findings, we determined that this episode of her chlamydial infection has been cured.

Reviewer 3 Report (New Reviewer)

In this case report entitled: Chlamydia peritonitis mimicking juvenile carcinomatous peritonitis diagnosed by exploratory laparoscopy: A Case Report, Nishida E. et al. reported a human case with Chlamydia trachomatis infection with gross pathology similar ovarian tumor and ascites. MRI examination suspected the patient had a hemorrhagic ovarian cyst and massive ascites. The CT showed the patient had massive ascites and peritoneal thickening. PET examination showed the patient had liver metastasis, para-aortic lymph node metastasis, and peritoneal dissemination. The laparoscopic examination surgery discovered that the patient had 200 mL of viscous yellow ascites, inflammatory peritoneum,  inflamed uterus, and swollen bilateral fallopian tubes. Following treatment with Azithromycin, the CA125 level went down to normal in 3 weeks, and WBC went up. The patient also had abdominal pain, which was controlled with non-steroidal anti-inflammatory drugs (NSAIDs)

Overall, this case report suggests C. trachomatis infection could cause pathology similar to an ovarian malignancy. Chlamydia peritonitis should be the differential diagnosis for cancer peritonitis in a juvenile patient with abnormal ascites. However, the case report did not discuss the issue of liver metastasis. The connection between CA125 level and chlamydia infection should be discussed..

The following are specific issues to be addressed:

  1. Figures 1 and 2 should be combined into one Figure. Change Fig 1 to 1A and fig 2 to 1B. Please circle the lesion and point it with arrows. 
  2. In Figure 3, please define the region where the arrow pointed to. What happened to that Liver metastasis after the surgery and treatment? 
  3. In Figure 4. the white lesion is not pointed by the arrow. 
  4. In Figure 7, please add dash line to indicate the normal ranges

Author Response

Dear Reviewer 3,

Thank you  very much for your comments. We modified the text accoringly.

--------------------------

Reviewer 3.

In this case report entitled: Chlamydia peritonitis mimicking juvenile carcinomatous peritonitis diagnosed by exploratory laparoscopy: A Case Report, Nishida E. et al. reported a human case with Chlamydia trachomatis infection with gross pathology similar ovarian tumor and ascites. MRI examination suspected the patient had a hemorrhagic ovarian cyst and massive ascites. The CT showed the patient had massive ascites and peritoneal thickening. PET examination showed the patient had liver metastasis, para-aortic lymph node metastasis, and peritoneal dissemination. The laparoscopic examination surgery discovered that the patient had 200 mL of viscous yellow ascites, inflammatory peritoneum,  inflamed uterus, and swollen bilateral fallopian tubes. Following treatment with Azithromycin, the CA125 level went down to normal in 3 weeks, and WBC went up. The patient also had abdominal pain, which was controlled with non-steroidal anti-inflammatory drugs (NSAIDs)

Overall, this case report suggests C. trachomatis infection could cause pathology similar to an ovarian malignancy. Chlamydia peritonitis should be the differential diagnosis for cancer peritonitis in a juvenile patient with abnormal ascites. However, the case report did not discuss the issue of liver metastasis. The connection between CA125 level and chlamydia infection should be discussed..

→Intraoperative findings, the liver surface, suspected to be a disseminated lesion from the preoperative imaging examination, was no abnormality founding, including adhesion to the peritoneum, so-called Fitz-Hugh-Curtis syndrome. We added comments to the intraoperative findings (lines 87-88). We add the discussion of the relationship between CA125 and Chlamydia infection. (lines 201-210. References [11].

The following are specific issues to be addressed:

  1. Figures 1 and 2 should be combined into one Figure. Change Fig 1 to 1A and fig 2 to 1B. Please circle the lesion and point it with arrows. 
  2. In Figure 3, please define the region where the arrow pointed to. What happened to that Liver metastasis after the surgery and treatment? 
  3. In Figure 4. the white lesion is not pointed by the arrow. 
  4. In Figure 7, please add dash line to indicate the normal ranges

→We fixed the figures according to your points. Also, we modified Figure 2 (a), (b), (c) to Figure 2 A, B, C

Round 2

Reviewer 2 Report (New Reviewer)

to authors:

References 8, 10, 12 are completely wrong and badly written 

to the Editor:

1. The manuscript is somewhat  improved, however PCR in urine should be always performed

2. References 8, 10, 12 are wrong and badly written. You should know that you spell the last name first and then the first initial of the first name. Please read well all the references and correct in: Contini C, Seraceni S, instead of Carlo C and Silva S, and also...Rotondo JC and not John CR

3. you mentioned references 8, 10 and 12, but you did not specify the context of the reference. What are they referred to? Please include in the text what they refer to.

Probably, azithromycin did give microbiological improvement (Chlamydia PCR test turned negative). Please, specify.

"We did not try to identify C. trachomatis in the uterine" . Did the authors mean urine??

Author Response

Dear Reviewer,

We wish to re-submit the manuscript in Pathogens, titled “Chlamydia peritonitis diagnosed by exploratory laparoscopy which mimicking juvenile carcinomatous peritonitis: A Case Report.” 

The manuscript has been rechecked and the necessary changes have been made in accordance with the reviewers’ suggestions document (wherein the reviewers’ comments are shown in blue italicized text). The responses to all comments have been prepared and are indicated with tracks on in the revised manuscript.

to authors:

References 8, 10, 12 are completely wrong and badly written 

→ I apologize for the lack of confirmation. I have corrected all.

to the Editor:

  1. The manuscript is somewhat improved, however PCR in urine should be always performed

→Chlamydia PCR test in the urine is rarely tested in routine gynecological practice. The reason is that Japan has universal health insurance, and it is financially difficult to submit multiple Chlamydia PCR tests. In gynecological practice, only a PCR test of cervical mucus is often sufficient for diagnosis. Chlamydia PCR tests in the urine are usually submitted only when there is a strong complaint of pain during urination or urethral itching, but this is not a standard test in gynecological practice.

  1. References 8, 10, 12 are wrong and badly written. You should know that you spell the last name first and then the first initial of the first name. Please read well all the references and correct in: Contini C, Seraceni S, instead of Carlo C and Silva S, and also...Rotondo JC and not John CR

→Sorry for the lack of confirmation. I have corrected all.

  1. you mentioned references 8, 10 and 12, but you did not specify the context of the reference. What are they referred to? Please include in the text what they refer to.

→ We have inserted a summary to clarify what I have quoted.; (line 186-188, line 213-233)

Probably, azithromycin did give microbiological improvement (Chlamydia PCR test turned negative). Please, specify.

→We added it to the last paragraph of the Discussion.

Finally, we administered azithromycin for the treatment after the Chlamydia infection was revealed by PCR test. From the clinical course and pathological examination results of this case, the patient was considered to have an acute to chronic Chlamydial infection. Although it is unclear how the patient's chlamydial infection was cured during the long clinical course, considering that oral azithromycin treatment resulted in negative chlamydial PCR, it can be inferred that it at least contributed to the microbiological improvement. (line 228-233)

"We did not try to identify C. trachomatis in the uterine" . Did the authors mean urine??

→I meant "We did not try to identify C. trachomatis in the urine."

Thank you very much for taking the time out of your precious time to leave us comments. We look forward to hearing from you regarding our submission.

Sincerely,

Kazunori Nagasaka

This manuscript is a resubmission of an earlier submission. The following is a list of the peer review reports and author responses from that submission.

Round 1

Reviewer 1 Report

This work is a case study that followed the examination of a young woman who presented symptoms associated with gynecologic cancer.  However, the individual was found to have had chlamydia infection, which probably lead to the associated symptoms. There should be an initial diagnosis of STDs before any invasive test for cancers be performed. 

Some of the paragraphs in the introduction require references, the statements made are hanging and should followed by references.

There is a need for a moderate grammatical correction, the text is easy to follow.

Figures 1 & 2 require arrows to signify what is being mentioned.

Figure 3 should be clearer.

Figure 5,6 & 7, should have axis titles, and the headings should be removed. The graphs should have the same fonts sizes. The figure title and legends can be written more appropriately.

What is the significance of adding the operation to these figures. Since the laproscopy was to determine the degree of inflammation and not to treat it.

There was a mentioned of a table, I am assuming that is a mistake since you have only graphs.

Was there a follow up after the treatment to find out if the chlamydia infection was negative and that the inflammation was not present?